# Development Perspective of Bioelectrocatalysis-Based Biosensors

**DOI:** 10.3390/s20174826

**Published:** 2020-08-26

**Authors:** Taiki Adachi, Yuki Kitazumi, Osamu Shirai, Kenji Kano

**Affiliations:** Division of Applied Life Sciences, Graduate School of Agriculture, Kyoto University, Sakyo, Kyoto 606-8502, Japan; adachi.taiki.62s@st.kyoto-u.ac.jp (T.A.); kitazumi.yuki.7u@kyoto-u.ac.jp (Y.K.); shirai.osamu.3x@kyoto-u.ac.jp (O.S.)

**Keywords:** current–potential curve, multi-enzymatic cascades, multianalyte detection, mass-transfer-controlled amperometric response, potentiometric coulometry

## Abstract

Bioelectrocatalysis provides the intrinsic catalytic functions of redox enzymes to nonspecific electrode reactions and is the most important and basic concept for electrochemical biosensors. This review starts by describing fundamental characteristics of bioelectrocatalytic reactions in mediated and direct electron transfer types from a theoretical viewpoint and summarizes amperometric biosensors based on multi-enzymatic cascades and for multianalyte detection. The review also introduces prospective aspects of two new concepts of biosensors: mass-transfer-controlled (pseudo)steady-state amperometry at microelectrodes with enhanced enzymatic activity without calibration curves and potentiometric coulometry at enzyme/mediator-immobilized biosensors for absolute determination.

## 1. Introduction

Electron transfer reactions such as photosynthesis, respiration and metabolisms play an important role in all living things. A huge variety of redox enzymes catalyze the oxidation and reduction of couples of two inherent substrates. Usually, the electrons are transferred between the two substrates through a cofactor(s) that is covalently or non-covalently bound to the redox enzymes. The most important ones are pyridine nucleotide coenzymes (NAD(P)(H)). The coenzymes shuttle back and forth between NAD(P)-dependent enzymes and solution and mediate hydride ion transfers (single step two-electron one-proton redox reactions) between two organic substrates in the biologic system without generating any intermediate radicals. On the other hand, there are a variety of metallic ion cofactors such as hemes, iron–sulfur clusters, copper ion, nickel ion and manganese ion, in redox enzymes. These metallic ion cofactors undergo single or multistep single-electron transfers (SETs). Flavin cofactors (FAD(H_2_) FMN(H_2_)) and quinone cofactors—including pyrroloquinoline quinone (PQQ)—can undergo both hydride ion transfers and SETs. Molybdopterin cofactors (Mocos) have similar properties. Therefore, flavin and quinone cofactors and Mocos have very important roles to mediate between the hydride ion transfer and the SET systems.

The coupling of redox enzymatic reactions with electrochemical reactions has received worldwide medical and scientific interests [1,2,3,4,5,6,7,8,9,10,11,12,13,14]. The coupled reaction is called bioelectrocatalysis. Since enzymatic reactions show very high performance in the selectivity, catalytic activity, uniformity and enormous chemical versatility, the coupling provides those redox–enzymatic characteristics to nonspecific electrode reactions. Therefore, several bioelectrochemical devices have been developed based on bioelectrocatalysis, such as biosensors [9,15], biologic fuel cells [16,17,18,19,20], bioelectrochemical reactors [21,22] and biosupercapacitors [23].

Most of redox–enzymatic reactions can be coupled with electrode reactions via redox mediators [3,24,25,26]. This reaction is called mediated electron transfer (MET)-type bioelectrocatalysis. Since electrode reactions are not hydride ion transfers, but single or multistep SETs, NAD(P)-dependent enzymatic reactions must be coupled to electrode reactions by using redox mediators that can transfer both hydride ion and single electrons, such as flavins, quinones (especially *o*-quinones) and phenothiazines (such as Meldola’s blue). For other redox enzymes (i.e., flavoenzymes, quinoenzymes, metal-containing enzymes), a large variety of redox compounds (that undergo SETs)—including metallic ion complexes (such ferrocenes, ferrocyanide, osmium complexes)—can be used as mediators.

On the other hand, it is known that some metal-containing enzymes and flavoenzymes can directly exchange electrons with electrodes in the absence of any mediators in the catalytic reaction. Such reactions are referred to as direct electron transfer (DET)-type bioelectrocatalysis [14,27,28,29,30,31]. In theory, DET-type bioelectrocatalysis can simplify the fabrication process of bioelectrodes and can minimize the thermodynamic overpotential in a coupled reaction. However, redox enzymes that can provide clear DET-type bioelectrocatalytic waves are limited in number, and DET-type reactions are very susceptible to the chemical properties and structure of the electrode surface.

On the basis of bioelectrocatalysis, amperometric biosensors are analytical devices used to detect specific target analytes (substrates) and have been widely used in various fields such as medical care, environmental monitoring, food safety and industrial bioprocess monitoring [12,13,32,33]. Generally, there are many compounds (analytes) in the target fluid. Therefore, multianalyte biosensors that may combine cost-effective and rapid analysis with reducing the volume of samples have been improved [34]. When one enzyme reaction cannot be effectively coupled with electrode reactions, multiple enzymes that work in a cascade mode are required. In this case, immobilization and colocalization of multiple enzymes are necessary for sensor fabrication, and various methods for immobilization and colocalization of multi-enzyme have been reported [35,36,37].

However, the response of an amperometric sensor depends on detection time, which limits practical application [4,9,11,12,13,26,32,33]. It is necessary for amperometric sensors to obtain steady-state currents. If one can use biosensors without calibration, it is easy to monitor the concentration of target analytes. In this sense, microelectrode detection was introduced to develop amperometric biosensors [38,39,40,41,42,43,44]. Nonlinear diffusion of the substrate at microelectrodes causes diffusion-controlled steady-state currents. In addition, microelectrode-type biosensors are suitable for miniaturization and reducing the volume of samples. On the other hand, coulometry that can determine the absolute quantity is an absolute analytical method. However, non-Faradaic background electricity (background integrated current during absolute electrolysis) becomes too large to be ignored, compared with Faradaic electricity to be measured at decreased concentrations of analytes. In order to overcome this issue, potentiometric coulometry based on MET-type bioelectrocatalysis has been proposed, in which the total electricity of an analyte in a small value of test solution is transferred to the mediator immobilized on an electrode, and the change in the redox state of the mediator is potentiometrically detected [45,46].

In this review, we describe fundamental concepts of MET- and DET-type bioelectrocatalytic reactions, practical use of biosensors such as multi-enzyme biosensor and multianalyte biosensor and prospective biosensors that can be utilized without calibration.

## 2. Fundamentals of Bioelectrocatalytic Sensors

Bioelectrocatalysis is the core reaction in electrochemical biosensors constructed with redox enzymes and electrodes. The effective coupling between the enzymatic and electrode reactions improves high performance for biosensors. In this section, the basic principle of the bioelectrocatalysis is introduced.

### 2.1. Theory of Steady-State Catalytic Currents in Met-Type Bioelectrocatalysis

#### 2.1.1. MET-Type Bioelectrocatalysis in Homogeneous System

In the MET-type bioelectrocatalysis (for the oxidation of a substrate (S) to a product (P)), the enzymatic reaction catalyzed by an enzyme (E) and the electrode reaction of a mediator (M) are coupled as follows:(1)$$S+\frac{n_{S}}{n_{M}}M_{O}\begin{array}{c} \mathrm{Enzyme}\\ \to \end{array}P+\frac{n_{S}}{n_{M}}M_{R},$$
and
(2)$$M_{R}\begin{array}{c} \mathrm{Electrode}\\ \to \end{array}M_{O}+n_{M}e^{-},$$
where *n*_X_ is the number of electrons of X, M_O_ and M_R_ are the oxidized and reduced forms of the mediator, respectively. In the presence of the enzyme, the mediator and the substrate in a quiescent solution, a 1D reaction–diffusion equation of the MET-type bioelectrocatalytic reaction under the steady-state conditions is expressed as follows:(3)$$\frac{\partial c_{M_{O}}}{\partial t}=D_{M}\frac{\partial^{2}c_{M_{O}}}{\partial x^{2}}-\frac{n_{S}}{n_{M}}\frac{k_{c}c_{E}}{1+\frac{K_{M}}{c_{M_{O}}}+\frac{K_{S}}{c_{S}}}=0,$$
where *c*_X_ is the concentration of X, *t* is the time, *x* is the distance from the electrode surface, *D*_M_ is the diffusion coefficient of the mediator, *k*_c_ is the catalytic constant, and *K*_X_ is the Michaelis constant for X, respectively. When the overpotential is high enough and the excess amount of S is present in solution (*c*_S_ >> *K*_S_), the limiting steady-state current (*i*_s,lim_) can be obtained and is expressed by solving Equation (3) [47]:(4)$$i_{s,\lim}=FA\sqrt{2n_{S}n_{M}D_{M}k_{c}K_{M}c_{E}\left\{\frac{c_{M}}{K_{M}}-\ln\left(1+\frac{c_{M}}{K_{M}}\right)\right\}},$$
where *F* is the Faraday constant, *A* is the electrode surface area, and $c_{M}\equiv c_{M_{O}}c_{M_{R}}$, respectively. Furthermore, Equation (4) is simplified in two limited cases of *c*_M_; when *c*_M_ << *K*_M_:(5)$$i_{s,\lim}=FAc_{M}\sqrt{n_{S}n_{M}D_{M}\frac{k_{c}}{K_{M}}c_{E}},$$
and when *c*_M_ >> *K*_M_:(6)$$i_{s,\lim}=FA\sqrt{2n_{S}n_{M}D_{M}k_{c}c_{E}c_{M}}.$$

#### 2.1.2. Reaction-Layer Model at Enzyme/Mediator-Immobilized Electrodes

Enzymes and mediators are often immobilized on the electrode surface for the application to biosensors. In such situations, an enzyme/mediator-co-immobilized layer and a reaction layer must be considered.

When the enzyme/mediator-co-immobilized layer is sufficiently smaller in thickness than the reaction layer (*L* << *µ*, *L* and *µ* being the thickness of immobilized and reaction layers, respectively), reaction (1) becomes a rate-determining step of the total reaction and the concentration polarization of the mediator and the substrate becomes negligible. Introducing a reaction layer theory [48] under the conditions of large overpotentials and *c*_S_ >> *K*_S_, *i*_s,lim_ is expressed as follows:(7)$$i_{s,\lim}=FAn_{S}D_{M}\frac{k_{c}c_{E}}{1+\frac{K_{M}}{c_{M_{O}}}}L.$$
when *L* >> *µ* or *L* ≈ *µ*, *i*_s,lim_ is expressed as follows:(8)$$i_{s,\lim}=FA\sqrt{2n_{S}n_{M}D_{M}k_{c}c_{E}B}\tanh\left\{\frac{L}{1+\frac{K_{M}}{c_{M}}}\sqrt{\frac{n_{S}}{n_{M}}\frac{k_{c}c_{E}}{2D_{M}K_{M}B}}\right\},$$
where $B=\frac{c_{M}}{K_{M}}-\ln\left(1+\frac{c_{M}}{K_{M}}\right)$. Equation (8) is simplified in two limited cases of *c*_M_, and *µ* is expressed as follows: When *c*_M_ << *K*_M_,
(9)$$i_{s,\lim}=FAc_{M}\sqrt{n_{S}n_{M}D_{M}\frac{k_{c}}{K_{M}}c_{E}}\tanh\left(\frac{L}{\mu }\right),$$
(10)$$\mu =\sqrt{\frac{n_{S}}{n_{M}}\frac{k_{c}c_{E}}{D_{M}K_{M}}};$$
and when *c*_M_ >> *K*_M_,
(11)$$i_{s,\lim}=FA\sqrt{2n_{S}n_{M}D_{M}k_{c}c_{E}c_{M}}\tanh\left(\frac{L}{\mu }\right),$$
(12)$$\mu =\sqrt{\frac{n_{S}}{n_{M}}\frac{k_{c}c_{E}}{2D_{M}c_{M}}}.$$

It is essential to optimize the value of *L*, in order to construct biosensors with large values of *i*_s,lim_/*A* using minimum amounts of enzymes and mediators.

#### 2.1.3. Serial Resistance Model

The steady-state MET-type bioelectrocatalysis can be explained based on a serial resistance model. In this model, we assume a set of series reactions (Figure 1): (1) mass transfer, (2) membrane permeation, (3) enzymatic reaction and (4) electrode reaction. The steady-state current (*i*_s_) can be expressed as follows:(13)$$\frac{1}{i_{s}}≅\frac{1}{i_{\mathrm{mt}}}+\frac{1}{i_{\mathrm{perm}}}+\frac{1}{i_{\mathrm{enz}}}+\frac{1}{i_{\mathrm{elec}}},$$
where *i*_mt_, *i*_perm_, *i*_enz_ and *i*_elec_ are the limiting steady-state currents controlled by mass-transfer, permeation, enzymatic reaction and electrode reaction, respectively. The value of *i*_mt_ is given by Levich equation at a rotating disk electrode (RDE) or by Equation (15) at a microdisk electrode:(14)$$i_{\mathrm{mt},\mathrm{RDE}}=\pm 0.620n_{S}FAD_{S}^{\frac{2}{3}}\nu^{-\frac{1}{6}}\omega^{\frac{1}{2}}c_{S},$$
(15)$$i_{\mathrm{mt},\mathrm{microdisk}}=\pm 4n_{S}FD_{S}rc_{S},$$
where *D*_S_ is the diffusion coefficient of the substrate, *ν* is a kinetic viscosity of the buffer solution, *ω* is an angular rotation rate of the RDE, and *r* is a radius of the microdisk electrode. The plus and minus signs in the current indicate the oxidation and reduction currents, respectively. The term *i*_perm_ can be expressed as follows:(16)$$i_{\mathrm{perm}}=\pm n_{S}FAP_{m}c_{S},$$
where *P*_m_ is a permeation coefficient of a membrane. The value of *i*_enz_ is given by a Michaelis–Menten-type equation in the presence of an excess amount of a mediator as follows:(17)$$i_{\mathrm{enz}}=\pm n_{S}FA\frac{k_{c}\Gamma_{E}}{1+\frac{K_{S}}{c_{S}}},$$
where *Γ*_E_ is the surface concentration of an enzyme. The value of *i*_elec_ is given by a Butler−Volmer-type equation as follows:(18)$$i_{\mathrm{elec}}=n_{S}FAk_{M}^{o}c_{S}\exp{\left\{\frac{n_{M,\mathrm{rds}}F}{RT}\left(E-E_{M,\mathrm{rds}}^{o\prime }\right)\right\}}^{1-\alpha_{M,\mathrm{rds}}},$$
where $k_{M}^{o}$ is the standard rate constant in the rate-determining step (rds) of the interfacial electron transfer of the mediator at the electrode, *R* is the gas constant, *T* is the absolute temperature, $n_{M,\mathrm{rds}}$ is the number of electrons in the rds of the mediator (generally $n_{M,\mathrm{rds}}$ = 1), *E* is the electrode potential, $E_{M,\mathrm{rds}}^{o\prime }$ is the formal potential of the rds of the mediator, and $\alpha_{M,\mathrm{rds}}$ is the transfer coefficient in the rds of the mediator.

Under the condition of *E* >> $E_{M,\mathrm{rds}}^{o\prime }$, *i*_elec_ becomes sufficiently larger than *i*_mt_, *i*_perm_, and *i*_enz_ to give a steady-state limiting current. On the other hand, *i*_perm_ depends on an outer membrane and *i*_enz_ is directly affected by $k_{c}\Gamma_{E}$ that often varies under measurement conditions such as temperature and pH. Thus, a diffusion-controlled steady-state condition (*i*_mt_ << *i*_perm_, *i*_enz_, and *i*_elec_) is ideal for amperometric biosensors, in which no calibration curve is required for determination.

### 2.2. Theory of Steady-State DET-Type Bioelectrocatalysis

In DET-type bioelectrocatalysis, an enzyme reacts with both a substrate and an electrode as follows:(19)$$S+E_{O}\begin{array}{c} \mathrm{Enzyme}\\ \to \end{array}P+E_{R},$$
(20)$$E_{R}\begin{array}{c} \mathrm{Electrode}\\ ⇄ \end{array}E_{O}+n_{E}e^{-},$$

In this situation, *i*_s_ can be expressed by the following equation based on the serial resistance model with a series of the mass-transfer process and the DET-type bioelectrocatalysis on the electrode surface (electro–enzyme reaction):(21)$$\frac{1}{i_{s}}=\frac{1}{i_{\mathrm{mt}}}+\frac{1}{i_{\mathrm{elec}-\mathrm{enz}}},$$
where *i*_mt_ is given by Equations (14) and (15) for RDE and microdisk electrode experiments, respectively. *i*_elec-enz_ (the electro–enzyme reaction-controlled current) can be expressed as follows [49]:(22)$$i_{\mathrm{elec}-\mathrm{enz}}=\pm \frac{n_{S}FAk_{c,\mathrm{DET}}\Gamma_{E}}{1+\frac{k_{c,\mathrm{DET}}}{k_{f}}+\frac{k_{b}}{k_{f}}},$$
where *k*_c,DET_ is the catalytic constant in DET-type bioelectrocatalysis, *k*_f_ and *k*_b_ are the interfacial electron transfer rate constants of the forward and backward reactions, respectively, which are given by the Butler–Volmer equations as follows: For oxidation,
(23)$$k_{f}=k_{E}^{o}\eta_{E}^{1-\alpha_{E,\mathrm{rds}}}$$
(24)$$k_{b}=k_{E}^{o}\eta_{E}^{-\alpha_{E,\mathrm{rds}}},$$
and for reduction,
(25)$$k_{f}=k_{E}^{o}\eta_{E}^{-\alpha_{E,\mathrm{rds}}}$$
(26)$$k_{b}=k_{E}^{o}\eta_{E}^{1-\alpha_{E,\mathrm{rds}}}$$
where $K_{E}^{o}$ is the standard rate constant in the rds of the interfacial electron transfer of the enzyme at the electrode, $\eta_{E}=\exp\left\{\frac{n_{E}^{\prime }F}{RT}\left(E-E_{E}^{o\prime }\right)\right\}$, $n_{E}^{\prime }$ is the number of electrons in the rds of the heterogeneous electron transfer of the enzyme (generally $n_{E}^{\prime }$ = 1), $E_{E}^{o\prime }$ is the formal potential in the rds of the enzyme, and $\alpha_{E,\mathrm{rds}}$ is the transfer coefficient in the rds of the enzyme. When *E* >> $E_{E}^{o\prime }$, *i*_elec-enz_ is limited to ± *n*_S_*FAk*_c,DET_*Γ*_E_.

In addition, $K_{E}^{o}$ decreases exponentially with an increase in the distance between the electrode surface and an electrode-active site of the enzyme (*d*) [50,51,52]. Thus, DET-type bioelectrocatalysis is often improved by using mesoporous electrode materials on which enzymes adsorb in increased probability of orientations favorable for DET-type reactions. Suitable modification of the electrode surface with chemical substances leads to electrostatic or specific attractive interaction between the electrode surface and the enzyme surface close to the electrode-active site [14,30].

### 2.3. Examples of MET/DET-Type Biosensors

The most popular enzymatic biosensors are those for self-monitoring of blood glucose (SMBG); blood glucose concentration being an important index in the treatment of diabetes. Various types of amperometric glucose biosensors have been reported [53,54,55,56,57,58,59]. SMBG sensors involve FAD-dependent glucose oxidase (GOD) [60,61], bacterial and fungal FAD-dependent glucose dehydrogenase (FAD-GDH) [62,63], PQQ-dependent soluble GDH (PQQ–sGDH) [64,65,66] and NAD-dependent GDH (NAD-GDH) [67]. Characteristics of the enzymes are summarized in a review [68].

Kakehi et al. reported a biofuel cell-type glucose biosensor using bacterial flavohemo-GDH complex containing a cytochrome subunit as well as an FAD subunit. The authors proposed that the enzyme proceeded DET-type bioelectrocatalysis at the cytochrome subunit and the open-circuit potential was used as a measure of the glucose concentration in the range of 0.5 mM to 6 mM [69].

Fructose biosensors are also useful in food analyses and diagnoses of kidney function. The sensors can be utilized for determining the inulin clearance, a difference of the intake and discharge of inulin filtered in the kidney, inulin being hydrolyzed into fructose and glucose by inulinase [70]. A membrane-bound flavohemo enzyme, d-fructose dehydrogenase (FDH), catalyzes two-electron oxidation of d-fructose, shows high activity of DET-type bioelectrocatalysis and is often mounted on fructose biosensors [71]. Both DET- and MET-type fructose biosensors are summarized in a review [72].

Cellobiose dehydrogenase (CDH) is useful for lactose biosensing [73]. CDH comprises an FAD-containing larger catalytic dehydrogenase domain and a heme *b*-containing smaller cytochrome domain that directly communicates with electrodes [73]. As a mimic of CDH, in addition, the cytochrome domain was introduced into some other flavoenzymes by protein engineering methods and the engineered enzymes realized DET-type bioelectrocatalysis. Ito et al. designed a cytochrome domain-linked fungal FAD-GDH and constructed a DET-type amperometric glucose biosensor [74].

## 3. Multi-Enzymatic Cascades

### 3.1. Diaphorase/NAD(P)^+^-Dependent Enzymes

NAD(P)^+^/NAD(P)H is a natural coenzyme involved in a large variety of redox enzymes. Usually, NAD(P)(H) shuttle back and forth between NAD(P)-dependent enzymes and solution to transfer the hydride ion. However, NAD(P) is an unfavorable mediator in MET-type reactions for NAD(P)-dependent enzymes, because the direct electrochemical reaction of NAD(P)(H) at electrodes requires high overpotentials, because NAD(P) is not a two-SET carrier, but a hydride ion carrier; SET characteristics are essential for electrode reactions. Therefore, other additional mediators, such as Meldola’s blue (MB) that can undergo both hydride ion transfer and SETs, were used as mediators. MB shows large values of the second-order reaction rate constant with NAD(P)H (hydride ion transfer) and also the heterogeneous electron transfer rate constant with electrodes (SET) [75]. Avramescu et al. constructed a d-lactate biosensor using NAD-dependent d-lactate dehydrogenase (d-LDH), NAD(H) and MB, with a detection limit of 0.05 mM, a linear range of 0.1–1 mM and a sensitivity of 1.2 μA cm^−2^ Mm^−1^ [76].

Furthermore, FMN-containing diaphorase (DI) that catalyzes a redox reaction between NAD(P)(H) and an artificial mediator can be utilized to build up more efficient mediated systems and was introduced in MET-type biosensors using NAD-dependent enzymes [77,78,79,80]. Takagi et al. analyzed bienzymatic MET-type bioelectrocatalysis of NAD-dependent l-lactate dehydrogenase (l-LDH) and DI using several mediators (Figure 2A) [77]. The bienzyme system realized an interconversion (two-way conversion) between l-lactate and pyruvate. Nikitina et al. reported an amperometric formaldehyde biosensor with NAD-dependent formaldehyde dehydrogenase, DI and an osmium redox polymer, with a detection limit of 32 μM, a linear range of 50–500 μM and a sensitivity of 2.2 μA cm^−2^ mM^−1^ [79].

On the other hand, Siritanaratkul et al. reported a DET-type interconversion of NADP^+^/NADPH catalyzed by ferredoxin-NADP^+^ reductase (FNR) [81]. They also demonstrated l-glutamate synthesis from 2-oxoglutarate and NH_4_^+^ by the coupled reactions of FNR and NADP-dependent glutamate dehydrogenase (GLDH) [81]. Multi-enzymatic biosensors using FNR have not yet been reported, but a DET-type NADP^+^/NADPH interconversion by FNR is potentially important in constructing NADP-dependent enzymatic biosensors without any additional mediators (Figure 2B).

### 3.2. Peroxidase/Oxidases

Several oxidases (ODs) were used as bioelectrocatalysts of 1st generation biosensors, which can detect and quantify target compounds by direct electrochemical oxidation of the enzymatically generated hydrogen peroxide (H_2_O_2_) to dioxygen (O_2_) [82,83,84]. However, direct oxidation of H_2_O_2_ at the electrode surface requires high overpotentials, and thus these biosensors are sensitive to interference by coexisting reductants [83,84].

In order to overcome this issue, H_2_O_2_ generated in an OD reaction was reductively detected with a help of horseradish peroxidase (HRP) that catalyzes two-electron reduction of H_2_O_2_ to H_2_O. The HRP reaction was frequently coupled with electrode reactions via MET-type bioelectrocatalysis. The two-enzyme reaction coupled electrode may be called OP-type biosensors, here. In OP-type biosensors, the following reactions proceed:(27)$$S+O_{2}\begin{array}{c} \mathrm{OD}\\ \to \end{array}P+H_{2}O_{2},$$
(28)$$H_{2}O_{2}+2H^{+}+2e^{-}\begin{array}{c} \mathrm{HRP}/\mathrm{electrode}\\ \to \end{array}2H_{2}O,$$

By introducing HRP, H_2_O_2_ detection at lower potentials was realized and the interference from other substances was reduced. The ODs mounted on OP-type biosensors reported so far are as follows: GOD [85,86], galactose oxidase (GalOD) [87], d- and l-amino acid oxidase [85,88,89], l-glutamate oxidase [90], amine oxidase [91], alcohol oxidase [85,91,92,93,94,95], urate oxidase [96], zinc superoxide dismutase [97], etc. Castillo et al. also investigated the bioelectrocatalytic characteristics of sweet potato peroxidase (SPP), and SPP-based OP-type biosensors showed higher performance than HRP-based ones [91].

In addition, multi-enzymatic biosensors were constructed by incorporating upstream enzymatic reactions in OP-type biosensors. Tkáč et al. constructed a glucose-non-interfering lactose biosensor by introducing β-galactosidase that hydrolyzes lactose to galactose and glucose, into the GalOD-HRP system [98]. Several groups reported cholesterol oxidase/cholesterol esterase/HRP co-immobilized biosensors, which enabled to monitor total cholesterol [99,100,101]. On the other hand, Nieh et al. reported a multi-enzymatic creatinine biosensor, as shown in Figure 3 [102].

Recently, it was found that HRP showed a DET-type bioelectrocatalytic activity at mesoporous carbon and nanostructured gold electrodes [103,104] and third generation OP-type biosensors using the DET-type reaction of HRP were reported. Xia et al. reported glucose and putrescine biosensors using GOD and putrescine oxidase, respectively [41,103]. In addition, Kawai et al. constructed an analytical model of an OP-type pyruvate biosensor based on the serial resistance concept [44].

## 4. Multianalyte Detection

Real samples are mixtures containing multiple components. In particular, biologic samples have complex compositions and interrelation among the compositions remains unclear. In addition, there are many diseases of which the specific biomarker has not been found out. Status of the human health is often evaluated on the basis of the balance of the compositions of the body fluid. Therefore, the multianalyte sensing and monitoring of biologic samples, that is, big data in human health attracted attention [105,106,107]. Similarly, multianalyte detection in environmental samples is also an important subject [108]. Electrochemical sensors are easily miniaturized and compatible with such multianalyte detection [109,110,111]. The development of printing technologies reduces the cost of the fabrication of bioelectrochemical sensors and realize a disposable sensor array system with a complicated structure. In this section, some notes in simultaneous multicomponent measurements are outlined.

### 4.1. Crosstalk among Amperometric Biosensors

Electrical circuits of the amperometric sensors are on the basis of potentiostat. In the simplest construction of potentiostat, the working electrode set to common potential [112]. The circuit functions to maintain the potential difference between the working and reference electrodes at the setting potential by current flowing. Therefore, it is basically impossible to employ multiple working electrodes and a single pair of reference and counter electrodes by multiple potentiostats. Moreover, when the multiple electrochemical systems locate in a sample solution, the unnoticed electric connection of the potentiostats (for example, sharing ground) possibly causes the current flow between the working electrodes. In order to avoid these problems in simultaneous measurements, it is necessary to employ a special apparatus (it is called multipotentiostat).

The desorption of enzymes and mediators from the electrode surface also becomes the origin of crosstalk among the sensing devices in biosensors for multianalyte detection. The immobilization of enzymes and mediators and employment of the permeable membrane are effective to reduce this type of crosstalk. In the disposable biosensor array, the design and layout of the sensors on the sensing device are important to prevent the cross-contamination within the measurement period.

### 4.2. Absolute and Relative Concentration

In order to reduce the damage of the subjects in sampling, discharged body fluids such as saliva, sweat, tear and urine are better samples than blood. The homeostasis of the blood is important in the living things, while the water contents in such discharged body fluids change readily with time, movement and uptake, etc. Therefore, determination of the relative concentration of the components to the internal standard becomes more important than the determination of the absolute concentration of the components in the body fluids. Here, the homeostasis of the internal standard will become problems because its concentration affects all over the measurements. Furthermore, the calibration of the individual sensors is a fatal problem because the calibration process of the sensor for the multiple-analyte is incompatible with the disposable sensors. Calibration of the sensors is required if the sensitivities of sensors are unstable. Therefore, the stability of the sensor is important in disposable sensors.

### 4.3. Examples of the Internal Standard in Body Fluids

The most widely used internal standard in the urine sample is creatinine. Creatinine is a metabolite of creatine in muscle. The produced creatinine transfers from blood to urine through the kidney. The steady-state production of creatinine in the human body is widely accepted. Therefore, in the blood of healthy people, the concentration of creatinine is practically constant, and the amount of creatinine transferred into urine is also constant. In urine analysis, creatinine is usually employed for calibration of other components. However, since renal dysfunction causes the variation of creatinine concentration in urine, creatinine is not a perfect standard [113].

The water content in sweat is easily varied. The suggested internal standards in sweat are chloride ion [114] and sodium ion [115]. Because it is difficult to incorporate those ions into the redox reaction, potentiometry is suitable to detect these ions.

In the metabolism of human bodies, l-lactate is a normal product. However, bacteria can produce both d- and l-lactates. Therefore, d-lactate in body fluids is a marker for bacterial infection [116]. In order to remove the dilution effect of body fluid, the most suitable reference material is l-lactate. Therefore, the simultaneous detection of lactate enantiomers is the most effective construction. In order to determine the d/l ratio of lactate in body fluid, the combination of the diffusion-limited amperometric d- and l-lactate sensors was reported based on MET-type bioelectrocatalysis of corresponding NAD-dependent enzymes and MB [43].

## 5. Prospective Biosensors without Calibration

Biosensors are fundamentally fragile, due to enzyme properties. The functions of biomaterials are easily suppressed by acid, base, oxidation, heat, dehydration and other external factors. Because the sensitivities of biosensors are generally labile to change, frequent calibrations of biosensors may be required to guarantee accuracy. In this section, prospective biosensors without calibration are introduced.

### 5.1. Significance of Mass-Transfer Controlling

As mentioned above, the current response of an amperometric biosensor is expressed as a serial combination of each reaction step. A simple case of the substrate reductive MET-type biosensing employing rotating disk electrode is considered here.

As mentioned in Section 2, the response of biosensors is affected by many physical quantities. However, the stability or reproducibility of these quantities is different from each other in the constructed biosensors. The features of the selectable or controllable values are summarized in Table 1. Generally, the electrode area (*A*) is easy to control except for nanoelectrodes. The formal potential of the mediator ($E_{M}^{o\prime }$) is possible to control by the selection of the mediator. Since the stability of redox mediators are limited due to oxygen and light damage, it is difficult to define the rigid redox state of mediators. Since the stability of the reference electrode is frequently poor, the value of *E* is unstable. The lack of uniformity of permeable membranes causes the poor reproducibility in *P*_m_. The rotating speed is easy to control. The most unstable component in the biosensor is enzymes. The value of $k_{c}\Gamma_{E}$ decreases by the denaturation of the enzyme with time. Therefore, the mass transfer process is the most stable and controllable in amperometric biosensors. The stabilization of biosensors must be achieved by setting the mass transfer process as the rate-determining step.

### 5.2. Bioelectrocatalysis at Microelectrodes

As described in the above (Section 2), the steady-state mass transfer from the solution to an electrode is realized by rotating electrodes or microelectrodes. However, since the rotating electrode requires a special apparatus, the application in the biosensors for real sample measurements seems to be difficult. Therefore, microelectrodes are more suitable for construction for amperometric biosensors. Another advantage of microelectrodes is a high signal–noise ratio because of the relatively high Faradaic current density against the charging current density.

Microelectrodes are classified into some types based on the symmetry of the shape, such as microsphere, microdisk, microcylinder and microband [112]. The microdisk electrode is the most available one. The time-dependence of the limited current of a microdisk electrode ($i_{\mathrm{md}}$) with a radius of *r* is given as follows [117],
(29)$$i_{\mathrm{md}}=4n_{S}FD_{S}c_{S}r\left\{0.7854+0.8862\sqrt{\tau_{d}}+0.2146\exp\left(-\frac{0.7823}{\sqrt{\tau_{d}}}\right)\right\},$$
where *τ*_d_ is the dimensionless time for the microdisk and is defined as follows:(30)$$\tau_{d}=\frac{4D_{S}t}{r^{2}}.$$

At the long-time limitation (*t*→∞), the current reaches a steady-state value given by Equation (15); the mass-transfer-limited current density at the microdisk electrode increases with a decrease in the radius of the electrode. On the other hand, the enzymatic reaction-limited current density would be constant at a constant surface density of the enzyme. Therefore, it would become difficult to satisfy the requirement for diffusion-controlled biosensors ($i_{\mathrm{enz}}$ >> $i_{\mathrm{mt},\mathrm{microdisk}}$) at microdisk electrodes with extremely small values of r. A similar situation is expected for other types of microelectrodes. In other words, most of the microelectrode-type biosensors are often controlled by the enzymatic kinetics (or the permeability of the outer membrane, if any), and therefore the response would be labile to change. However, if one can realize extremely fast enzymatic reactions on microelectrodes, one may satisfy the condition: $i_{\mathrm{enz}}$ >> $i_{\mathrm{mt},\mathrm{microdisk}}$.

The simplest case is extremely fast DET-type bioelectrocatalysis at a microdisk electrode. In this situation, the substrate concentration at the microelectrode surface is regarded as zero, that is, diffusion-controlled limiting current conditions are realized. Some of multicopper-oxidases (MCOs) reduce O_2_ to water with sufficiently high activity at the surface of mesoporous electrodes by DET-type bioelectrocatalysis. Mass-transfer-controlled DET-type bioelectrocatalysis was realized at MCO-modified porous gold microdisk electrodes [118]. The porous electrode was employed to increase the effective surface area and the probability of orientations suitable for DET-type bioelectrocatalysis thanks to curvature effects [19,119,120]. Figure 4A shows voltammograms recorded at a Cu-efflux oxidase-modified porous gold microelectrode (solid line) with and (dotted line) without oxygen. The sigmoidal curve indicates the DET-type bioelectrocatalysis of O_2_ reduction. A clear potential-independent limiting current was observed. The limiting current linearly increased with the bulk concentration of O_2_, as shown in Figure 4B. The solid line in Figure 4B shows the theoretical sensitivity calculated with Equation (15) using the literature value of *D*_S_ and the microscopically measured *r*. The agreement between the experimental and theoretical sensitivities verifies that the response of the constructed O_2_ sensor is truly controlled by the mass transfer of O_2_. Since the sensitivity of the sensor is independent of the activity of CueO, the sensor shows excellent reproducibility and stability at a given temperature.

The situation of MET-type bioelectrocatalysis at a microelectrode is more complicated than that of DET-type bioelectrocatalysis due to the diffusion of the mediator. However, when a large amount of active enzymes exist in the system, it is possible to realize the substrate–diffusion-controlled MET-type bioelectrocatalysis at a microdisk electrode. According to this concept, a MET-type glucose sensor was constructed using a microdisk electrode, 1,2-benzoquinone and FAD-GDH [39]. The current response quickly reached a steady-state (Figure 5). The linear range of the sensor was from 0 mM to 3 mM of glucose while the concentration of the mediator was 1 mM in the system (inset in Figure 5). Obviously, the linear range was beyond the endpoint situation. Moreover, the current response of the sensor agreed with the diffusion-controlled value of glucose.

The situation where the MET-type bioelectrocatalysis occurs around a microelectrode was simulated by the finite element method [40]. The concentration profiles of the substrate, reduced form of the mediator, and reduced enzyme are given in Figure 6 under the conditions that the enzymatic reaction is extremely fast. While the concentration gradient of the substrate spreads hemispherically, the concentration gradient of the mediator is located only in the vicinity of the electrode surface. The concentration profile of the enzyme clearly shows that the enzymatic reaction occurs only in the thin region (as a reaction plane) where the enzyme concentration gradient exists. The flux of the substrate is quickly converted to that of the mediator in the thin region. Therefore, the surface of the thin region plays the role of a virtual electrode that selectively reacts with the substrate. The increase in the substrate concentration leads to a decrease in the distance of the thin region and then increases the current density.

### 5.3. Pseudo-Steady-State Response

Although microdisk electrodes provide clear steady-state current, the current is quite small. In order to increase the current, microdisk electrode array without overlapping of the diffusional concentration gradients (scattered microdisk electrode array) is one of the solutions [38,121,122,123]. However, the fabrication of such a scattered microdisk electrode array is a technically challenging issue. Another solution to increase the current may be the employment of microband electrodes. Ultrathin ring electrodes can be considered as rod electrodes [124,125]. The time-dependence of the limiting current at a microband electrode ($i_{\mathrm{mb}}$) with a length of *l* and a thickness of *w* is given by [126]:(31)$$i_{\mathrm{mb}}=n_{S}FD_{S}c_{S}l\left[\frac{\pi \exp\left(-\frac{2}{5}\sqrt{\pi \tau_{b}}\right)}{4\sqrt{\pi \tau_{b}}}+\frac{\pi }{\ln\left\{64\tau_{b}\exp\left(-0.5772156\right)+\exp\left(\frac{5}{3}\right)\right\}}\right],$$
where *τ*_b_ is the nondimensional time for the band electrode and is defined as follows,
(32)$$\tau_{b}=\frac{D_{S}t}{w^{2}}.$$

According to Equation (31), $i_{\mathrm{mb}}$ easily magnifies with an increase in *l*. On the other hand, $i_{\mathrm{mb}}$ becomes 0 at *t*→∞. Therefore, the exact analysis of $i_{\mathrm{mb}}$ based on the steady-state current is not possible. However, the decay of $i_{\mathrm{mb}}$ is so slow that the current is practically indistinguishable from the so-called steady-state current. In actual electrochemical measurements, a quasi-steady-state response is acceptable for practical use. For example, MET-type bioelectrocatalysis of FAD-GDH at microband electrodes provided the values very close to the diffusion-limited response [42].

Since the current density at a microelectrode is quite large, the mass-transfer-limited bioelectrocatalysis is fundamentally difficult. Therefore, a relatively large microelectrode (*r* = 1.5 mm) is promising to realize the mass-transfer-limited bioelectrocatalysis. In Equation (29), time is a dimensionless quantity and the scale is determined by *r*. Therefore, Equation (29) is more suitable to define the limiting current decay at a disk electrode than the Cottrell equation. The current decay given by Equation (29) is slower than that of the Cottrell equation. Even if the electrode radius is few mm, the limiting current reaches quasi-steady-state values. The quasi-steady-state characteristics of a disk electrode (*r* = 1.5 mm) realized diffusion-controlled lactate biosensing [43].

### 5.4. Potentiometric Coulometry

Coulometry is one of the absolute quantification methods and the most accurate electroanalytical methods in theory. In coulometry, the charge (electricity) due to the objective redox reaction is measured. Coulometry coupled with bioelectrocatalysis is a familiar technique [127,128,129,130,131]. However, small Faradaic current compared with the non-Faradaic current at porous electrodes causes a decrease in the accuracy of bioelectrocatalytic coulometry.

On the other hand, potentiometry is expected to avoid the effect of non-Faradaic current. When redox mediators exist in the solution, the redox enzymatic reaction with the mediator changes the ratio of the oxidized and reduced forms of the mediator. The change of the ratio changes the equilibrated solution potential based on the Nernst equation. When the redox mediator is immobilized at the electrode surface in a thin layer, the change of the ratio of the oxidized and reduced forms of the mediator (*Γ*_O_/*Γ*_R_) changes the equilibrated electrode potential (*E*). If the total amount of immobilized mediator (*AΓ*_T_) is constant, the accumulated charge (*Q*_S_) is evaluated directly from the initial value and equilibrated value of *E* (*E*_i_ and *E*_f_, respectively) as follows:(33)$$Qs=\frac{-n_{M}FA\Gamma_{T}}{1+\exp\left[\frac{n_{M}F}{RT}\left(E_{f}-E^{\circ }\prime \right)\right]}+\frac{n_{M}FA\Gamma_{T}}{1+\exp\left[\frac{n_{M}F}{RT}\left(E_{i}-E^{\circ }\prime \right)\right]}.$$

However, in the case of osmium complex polymer as a mediator immobilized at the electrode surface, the relationship between *E* and *Γ*_O_/*Γ*_R_ was deviated from the Nernst equation [45]. The most possible cause is strong electrostatic interactions between the redox sites in the polymer. An increase in the ionic strength in the medium will decrease the electrostatic interaction between the redox species at the electrode surface. In order to increase the ionic strength, a redox active thin liquid film containing a highly concentrated electrolyte solution was fabricated on the electrode surface [46]. Figure 7 shows the construction of the liquid film-modified electrode. The components of the thin liquid film were a hydrophobic ionic liquid (1-ethyl-3-methylimidazolium bis(nonafluorobutanesulfonyl)imide), organic medium (dibutyl phthalate) and a redox mediator (ferrocene). Ferrocene is enzymatically oxidized at the surface of the liquid film by H_2_O_2_ with peroxidase (POD). Since the electrostatic interaction was effectively suppressed, the liquid film-modified electrode played as a reversible and ideal surface-confined system. Since the value of *AΓ*_T_ is controlled in the fabrication of the modified electrode, the value of *Q*_S_ by the bioelectrocatalysis can be estimated from Equation (33) without any calibration.

The sensitivity of the potentiometric coulometry could be regulated on the basis of the value of *AΓ*_T_. Since no current flows across the system, the interference due to the non-Faradaic processes could be eliminated by potentiometric coulometry.

## 6. Conclusions

Over the past few decades in the field of bioelectrochemistry, several attempts have been made to utilize redox enzymes as electrocatalysts and to develop novel bioelectrochemical systems. By comparison with inorganic catalysts, redox enzymes have distinctive characteristics such as high activity, extremely large size, identity by regeneration, uniformity, versatility and fragility. In this review, we provide an overview and additional insights and discuss the recent progress on the practical use of amperometric biosensors such as multi-enzyme biosensors, multianalyte biosensors. The characteristics of each type of bioelectrocatalysis are summarized in Table 2. Prospective diffusion-controlled biosensors based on DET- and MET-type reactions and potentiometric coulometry based on MET-type reaction may be utilized without calibration. All these efforts may be useful for constructing bioelectrochemical sensors for practical use.

## Figures and Tables

**Figure 1 sensors-20-04826-f001:**
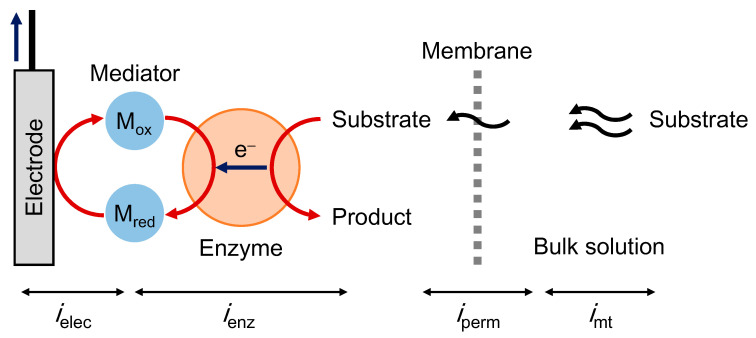
Schematic view of mediated electron transfer (MET)-type bioelectrocatalysis based on a serial resistance model.

**Figure 2 sensors-20-04826-f002:**
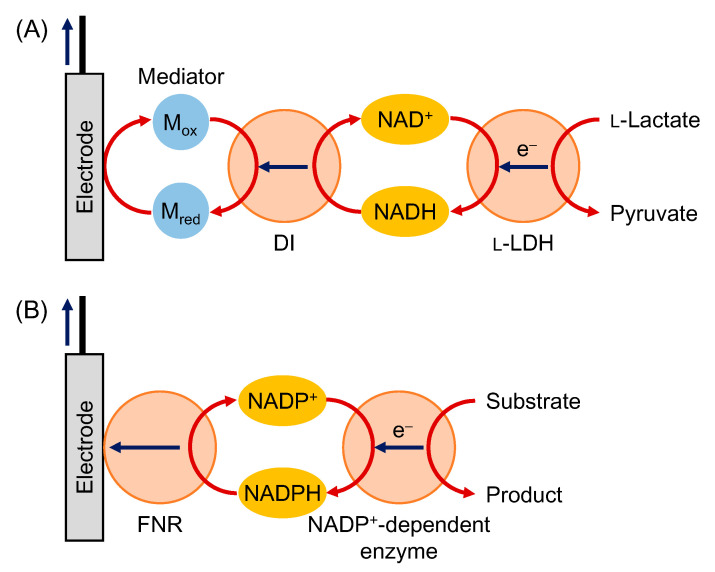
(**A**) Schematic view of MET-type bioelectrocatalysis of l-lactate dehydrogenase (l-LDH) and FMN-containing diaphorase (DI); (**B**) schematic view of bienzymatic bioelectrocatalysis using an NADP^+^-dependent enzyme and ferredoxin-NADP^+^ reductase (FNR).

**Figure 3 sensors-20-04826-f003:**
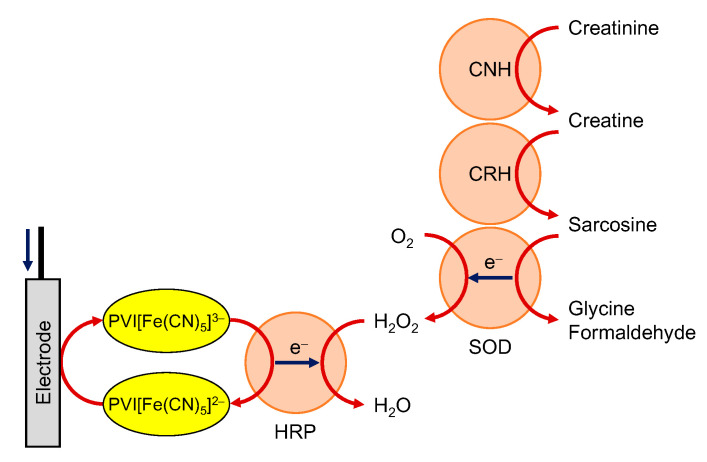
Schematic view of multi-enzymatic creatinine biosensor. CNH, CRH, SOD and PVI[Fe(CN)_5_] ^2−/3−^ indicate creatinine amidohydrolase, creatine amidohydrolase, sarcosine oxidase and pentacyanoferrate-bound poly(1-vinylimidazole), respectively.

**Figure 4 sensors-20-04826-f004:**
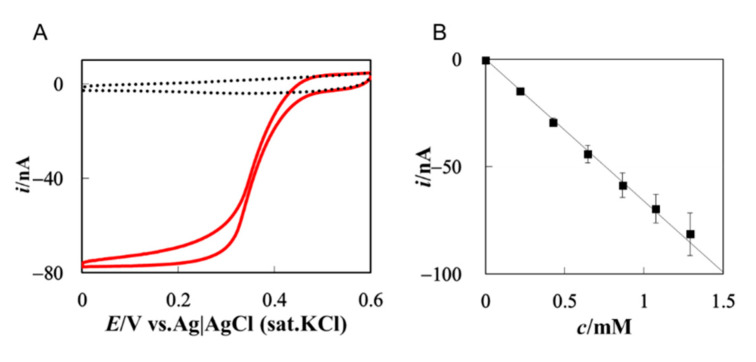
(**A**) Cyclic voltammogram of CueO-modified porous gold microdisk electrode recorded with *r* = 20 μm (solid line) with and (dotted line) without oxygen; (**B**) (squares) experimental and (solid line) theoretical calibration curve for the oxygen biosensor. Reprinted from ref. [118], Copyright (2020), with permission from Elsevier.

**Figure 5 sensors-20-04826-f005:**
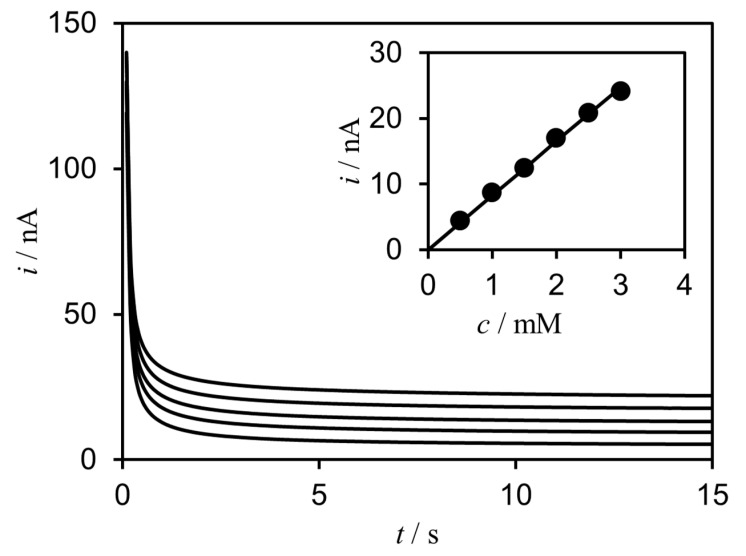
Chronoamperometric response for glucose oxidation in a buffer solution containing 0.21-mM FAD-dependent glucose dehydrogenase (FAD-GDH) and 1-mM 1,2-benzoquinone at a microelectrode with diameter of 50 μm. Inset shows the calibration curve based on the current at 10 s. Reprinted with permission from ref. [39], Copyright (2013) Japan Society for Analytical Chemistry.

**Figure 6 sensors-20-04826-f006:**
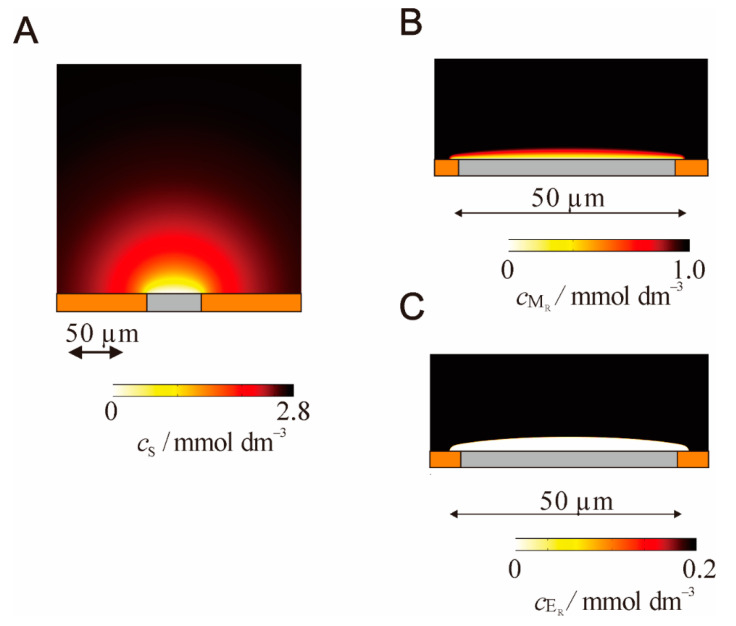
Concentration profiles of (**A**) substrate, (**B**) reduced mediator and (**C**) reduced form of enzyme around an microdisk electrode with *r* = 25 μm at a substrate concentration of 4 mmol dm^−3^, a mediator concentration of 1 mmol dm^−3^, an enzyme concentration of 0.2 mmol dm^−3^. Profiles calculated for the situation of 20 s after the potential step at the limiting current conditions. Reprinted from ref. [40] with permission from the PCCP Owner Societies.

**Figure 7 sensors-20-04826-f007:**
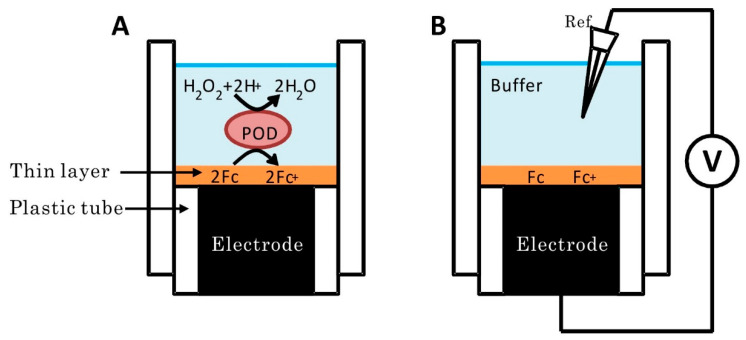
Schematic illustrations of (**A**) accumulation process and (**B**) potentiometric measurement of a liquid-film-modified electrode. Reprinted from ref. [46], Copyright (2015), with permission from Elsevier.

**Table 1 sensors-20-04826-t001:** Physical quantities affected to the amperometric biosensing.

Physical Quantities	Stability	Reproducibility	Controllability
Surface area of electrode (*A*)	Good	good	good
Standard redox potential ($E_{M}^{o\prime })$	good	good	poor
Concentration of mediator (*c*_M_)	poor	good	good
Electrode potential (*E*)	poor	poor	good
Permeability of membrane (*P*_m_)	good	poor	poor
Rotating speed (*ω*)	good	good	good
Enzyme activity ($k_{c}\Gamma_{E})$	poor	poor	poor

**Table 2 sensors-20-04826-t002:** Advantages and disadvantages of the bioelectrocatalytic systems for biosensing.

Type of Bioelectrocatalysis	Advantage	Disadvantage
MET-type	Easy coupling of enzyme reaction High loading of enzyme and mediator per projected area	Leakage of mediator and/or enzyme Stability of mediator Low thermodynamic efficiency
DET-type	Crosstalk-free High thermodynamic efficiency	Limited amounts of effective enzyme per projected area Limited number of enzymes Interference from strongly adsorbing substances
Multi-enzymatic cascade	Flexibility in sensor design	Instability due to series reactions

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
