# Peer review of "Development Perspective of Bioelectrocatalysis-Based Biosensors"

_sensors, 2020, doi:10.3390/s20174826_

Round 1
Reviewer 1 Report
The paper provides a comprehensive review of various bioelectrocatalytic reactions and their practical biosensing applications.
To make the presentation easier to follow, I suggest adding a table that summarize the features, advantages, disadvantages, and practical applications of different types of bioelectrocatalysis sensors: MET-based, DET-based, multi-enzymatic cascades, multi-analyte detection etc. This way would make the paper easier to follow.
I would also suggest the authors to provide more insights regarding the general designing principles for bio-electrocatalytic sensors and the further practical implications of the various reactions discussed. This would make the review article more valuable.
Since this is a review article, I suggest adding additional discussion on the comparison between enzymatic biosensors and non-enzymatic biosensors such as nanotubes-based sensors discussed in the three papers below. This would make the paper more interesting to the general audience.
- Appl. Phys. Lett. 102, 183113 (2013)
- Nanoscale 9 (4), 1687-1698
- Nano Lett. 2013, 13, 2, 625–631
Finally, after equations (1) and (2), please mention M_o means oxidized mediator and M_r means reduced mediator. Moreover, in equations (14)-(17), why there is a +/- sign in front of the current expression?
Author Response
The paper provides a comprehensive review of various bioelectrocatalytic reactions and their practical biosensing applications.
To make the presentation easier to follow, I suggest adding a table that summarize the features, advantages, disadvantages, and practical applications of different types of bioelectrocatalysis sensors: MET-based, DET-based, multi-enzymatic cascades, multi-analyte detection etc. This way would make the paper easier to follow.
>>A table that summarizes some properties of the amperometric biosensor systems was added in Conclusion.
I would also suggest the authors to provide more insights regarding the general designing principles for bio-electrocatalytic sensors and the further practical implications of the various reactions discussed. This would make the review article more valuable.
>>The basic construction of the electrochemical biosensors is briefly introduced in the beginning of section 2.
Since this is a review article, I suggest adding additional discussion on the comparison between enzymatic biosensors and non-enzymatic biosensors such as nanotubes-based sensors discussed in the three papers below. This would make the paper more interesting to the general audience.
- Appl. Phys. Lett. 102, 183113 (2013)
- Nanoscale 9 (4), 1687-1698
- Nano Lett. 2013, 13, 2, 625–631
>> The authors appreciate the reviewer's suggestion. However, the sensors in the references are constructed based on FET-type CNT works. The concept of the FET-type sensor is very important in the discussion of biosensors, but is different from that of the present issues in our review. Therefore, we could not accept the comment on the FET-type sensor for this review.
Finally, after equations (1) and (2), please mention M_o means oxidized mediator and M_r means reduced mediator. Moreover, in equations (14)-(17), why there is a +/- sign in front of the current expression?
>> The expression about MO and MR is added in the text. +/- signs in the current indicate the oxidation and reduction currents, respectively. The explanation is added to the text.
Reviewer 2 Report
The authors presented a review on the fundamental characteristics of bioelectrocatalytic reactions and prospective aspects of two new concepts of biosensors based on mass‐transfer‐controlled (pseudo)steady‐state amperometry and potentiometric coulometry. This manuscript shows a very well-organized work, the results from the available literature being comprehensively described. Therefore, I recommend its publication in Sensors.
I suggest to write “Controllability” in Table 1 and to detail the names of all physical parameters.
Also, in Figure 7 please write “Plastic tube”.
Author Response
The authors presented a review on the fundamental characteristics of bioelectrocatalytic reactions and prospective aspects of two new concepts of biosensors based on mass‐transfer‐controlled (pseudo)steady‐state amperometry and potentiometric coulometry. This manuscript shows a very well-organized work, the results from the available literature being comprehensively described. Therefore, I recommend its publication in Sensors.
I suggest to write “Controllability” in Table 1 and to detail the names of all physical parameters.
Also, in Figure 7 please write “Plastic tube”.
>> Table 1 and Figure 7 were revised.
Reviewer 3 Report
This MS, “Adachi et al., sensors-887994”, is an excellent review on development perspective of bioelectrocatalysis-based biosensors. The structure of the MS is good; it covers the most important topics available in the literature. The English of the MS is good. I did not find any mistake, not even a single mistype. I definitely suggest publishing it basically in the present form.
I have only one comment to the authors.
I disagree with the statement
“… NAD(P) is not an electron carrier but a hydride ion carrier…”. (Ls: 201-202)
In my opinion hydride ion is the frame for carrying two electrons.
Author Response
This MS, “Adachi et al., sensors-887994”, is an excellent review on development perspective of bioelectrocatalysis-based biosensors. The structure of the MS is good; it covers the most important topics available in the literature. The English of the MS is good. I did not find any mistake, not even a single mistype. I definitely suggest publishing it basically in the present form.
I have only one comment to the authors.
I disagree with the statement
“… NAD(P) is not an electron carrier but a hydride ion carrier…”. (Ls: 201-202)
In my opinion hydride ion is the frame for carrying two electrons.
>> The description has been revised in Introduction and Section 3.1.